# Nanoscale binding site localization by molecular distance estimation on native cell surfaces using topological image averaging

Vibha Kumra Ahnlide[1], Johannes Kumra Ahnlide[1], Sebastian Wrighton[1], Jason P Beech[2], Pontus Nordenfelt[1]*

[1]Division of Infection Medicine, Department of Clinical Sciences, Faculty of Medicine, Lund University, Lund, Sweden; [2]Division of Solid State Physics, Department of Physics, Lund University, Lund, Sweden

**Abstract** Antibody binding to cell surface proteins plays a crucial role in immunity, and the location of an epitope can altogether determine the immunological outcome of a host-target interaction. Techniques available today for epitope identification are costly, time-consuming, and unsuited for high-throughput analysis. Fast and efficient screening of epitope location can be useful for the development of therapeutic monoclonal antibodies and vaccines. Cellular morphology typically varies, and antibodies often bind heterogeneously across a cell surface, making traditional particle-averaging strategies challenging for accurate native antibody localization. In the present work, we have developed a method, SiteLoc, for imaging-based molecular localization on cellular surface proteins. Nanometer-scale resolution is achieved through localization in one dimension, namely, the distance from a bound ligand to a reference surface. This is done by using topological image averaging. Our results show that this method is well suited for antibody binding site measurements on native cell surface morphology and that it can be applied to other molecular distance estimations as well.

*For correspondence: pontus.nordenfelt@med.lu.se

Competing interest: The authors declare that no competing interests exist.

## Editor's evaluation

You have developed and validated a new method for measurement of nanoscale height of macro-molecules that can be non-uniformly distributed on irregular surfaces. Such samples are common in biology, which will make this a valuable approach to achieve super-resolution results for samples where this would have not been possible previously.

## Introduction

The location of protein binding sites on cellular surfaces can have wide-ranging implications for various cellular processes, such as immune signalling, cell adhesion, cell migration, and phagocytosis. Antibody binding to pathogen surface proteins plays a crucial role in immunity (*Lu et al., 2018*), and epitope localization affects diverse immunological outcomes, such as antibody neutralizing ability (*Law et al., 2008*; *Kwong et al., 2013*; *Salinas et al., 2019*; *Caoili, 2014*; *Ivanyi, 2014*) or autoreactivity (*Caoili, 2014*; *Cunningham, 2014*). Even though monoclonal antibody (mAb) treatments have proven to be successful for a wide spectrum of diseases, there are very few monoclonal antibodies clinically available for the treatment of bacterial infections (*Motley et al., 2019*). Developing therapeutic mAbs against infections may require large-scale screening of highly conserved and functional

**eLife digest** Antibodies play a key role in the immune system. These proteins stick to harmful substances, such as bacteria and other disease-causing pathogens, marking them for destruction or blocking their attack. Antibodies are highly selective, and this ability has been used to target particular molecules in research, diagnostics and therapies.

Typically, antibodies need to stick to a particular segment, or 'epitope', on the surface of a cell in order to trigger an immune response. Knowing where these regions are can help explain how these immune proteins work and aid the development of more effective drugs and diagnostic tools.

One way to identify these sites is to measure the nano-distance between antibodies and other features on the cell surface. To do this, researchers take multiple images of the cell the antibody is attached to using light microscopy. Various statistical methods are then applied to create an 'average image' that has a higher resolution and can therefore be used to measure the distance between these two points more accurately. While this approach works on fixed shapes, like a perfect circle, it cannot handle human cells and bacteria which are less uniform and have more complex surfaces.

Here, Kumra Ahnlide et al. have developed a new method called SiteLoc which can overcome this barrier. The method involves two fluorescent probes: one attached to a specific site on the cell's surface, and the other to the antibody or another molecule of interest. These two probes emit different colours when imaged with a fluorescent microscope. To cope with objects that have uneven surfaces, such as cells and bacteria, the two signals are transformed to 'follow' the same geometrical shape. The relative distance between them is then measured using statistical methods. Using this approach, Kumra Ahnlide et al. were able to identify epitopes on a bacterium, and measure distances on the surface of human red blood cells.

The SiteLoc system could make it easier to develop antibody-based treatments and diagnostic tools. Furthermore, it could also be beneficial to the wider research community who could use it to probe other questions that require measuring nanoscale distances.

---

epitopes (*Martín-Galiano and McConnell, 2019*). Antibody epitopes are typically identified using crystallography, mutagenesis, or crosslinking coupled mass spectrometry (*Abbott et al., 2014*). As of today, these methods are costly, time-consuming, and unsuitable for high-throughput analysis. Fast and efficient screening of epitope locations can be useful for the development of therapeutic mAbs and vaccines.

While averaging methods for nanometer-scale localization on microscopy data exist, these do not work for binding sites on proteins with heterogeneous expression on the cellular surface. Moreover, they are designed for identical (*Szymborska et al., 2013*; *Laine et al., 2015*) or spherical (*Son et al., 2020*) particles and are thereby not fitted for unaltered cells due to the existing biological variation of surface morphology.

In the present work, we have developed an imaging-based method for localization of binding sites with nanometer-scale precision. The location of a binding site is determined by calculating the average distance between the ligand channel and a reference signal channel using the averaged fluorescent signal along a cell contour. To assess the performance of this method, we analyse simulated images, demonstrating high accuracy for varying surface expression and morphology. By measuring the height of DNA probes on erythrocyte surfaces, we show that our method can accurately estimate molecular heights on spherical and non-spherical cells. Our presented results with structured illumination microscopy (SIM) data achieve an ~10 nm precision certainty. We apply the site localization method to determine known binding site locales on M protein that is unevenly expressed on the bacterial surface of *Streptococcus pyogenes*. Additionally, our method yields viable results when we perform site localization on widefield images with and without deconvolution. To sum up, our site localization method enables rapid determination of binding sites without the need for synthetic modification of cell surface morphology.

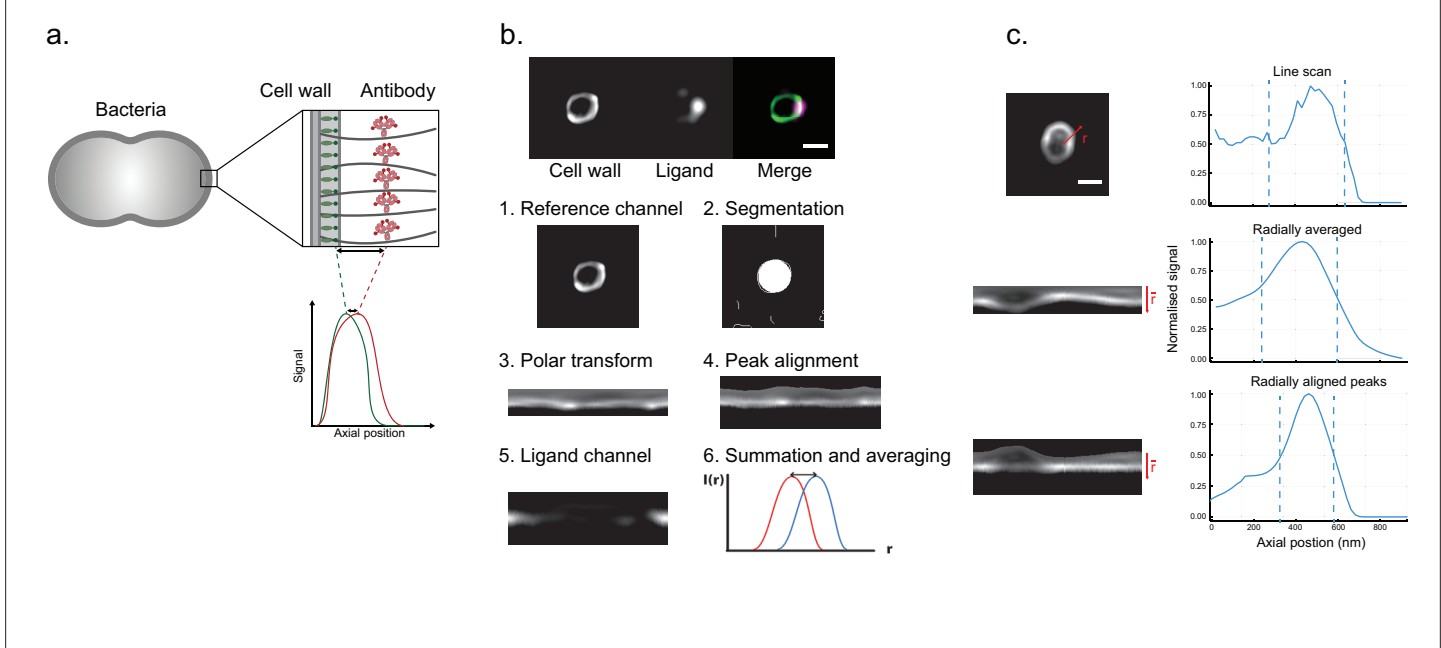

**Figure 1.** Binding site localization is based on resolving the distance between bound ligands and a reference surface. (**a**) Site localization on bacterial surfaces. The illustration exemplifies a measurement with antibody-coated bacteria. Antibodies with red fluorescent dyes are bound to bacterial surface proteins. A bacterial cell wall is labelled with a green fluorescent dye. High-resolution images of single bacteria are acquired in the focal plane, and the fluorescent peak signal is averaged along the bacterial contour. The distance d is determined by resolving the difference between the fluorescent signal peaks. (**b**) Analysis pipeline for the site localization method. Structured illumination microscopy (SIM) images are shown at the top. *S. pyogenes* strain SF370 has been fixed, stained with Alexa Fluor 488-conjugated wheat germ agglutinin (WGA), and coated with antibody Fc fragments. The antibody fragments were stained with Fc-specific Alexa Fluor 647-conjugated F(ab')₂ fragments. The scale bar is 500 nm. The aim is to locate the antibody binding site by calculating the average distance between the antibody channel and a reference signal channel. (1) A raw image in the reference channel is shown. (2) Bacteria are identified by fitting circles to an edge-detection-processed image. (3) The data in each circle mask is isolated and transformed to polar coordinates. (4) An alignment of the reference position is performed by identifying peaks at each radial position. (5) The spatially corresponding data in the antibody channel is extracted. (6) The radially aligned peaks are then averaged. The peak distance between these intensity profiles should then correspond to the distance between the binding site and the bacterial peptidoglycan layer. (**c**) Representation of improved SNR by peak alignment for non-spherical particles Intensity profiles (right) for the reference channel is shown together with their respective image data (left). The signal along a single line for an oval-shaped bacterium is shown at the top. The intensity profile of a radial average is shown in the middle. An improved SNR is seen as a peak alignment is performed (bottom).

## Results

### Principles of site localization method

Binding site localization is based on resolving the axial distance between bound ligands and a reference surface. This is done by implementing an averaging method on high-resolution images. With this method, nanometer-scale precision is achieved by dimension reduction of surface data within an image plane.

A site localization measurement on antibody-coated bacteria is exemplified in *Figure 1*. Fluorescently conjugated wheat germ agglutinin (WGA) that binds bacterial peptidoglycan is used here for the reference region, and antibodies that bind to bacterial M protein are labelled with another fluorescent dye for imaging in a separate channel (*Figure 1a*). The example shows how our method handles the uneven distribution of M protein on bacterial surfaces (*Figure 1b*). The pipeline for the site localization method is shown in *Figure 1b*. For each image, bacteria are identified in the reference channel by fitting circles to an edge-detection-processed image. Each circle is isolated as a slightly larger mask and the data in the circle is polar transformed. An alignment of the reference position is performed by identifying peaks to each intensity profile. This way, the axial positions are normalized to the actual surface signal. The spatially corresponding data in the antibody channel is processed in the same manner, and the distance between the antibody and WGA signal is resolved through the averaging of multiple acquisitions of the same field of view. Altogether, a relative binding site is

determined by resolving the distance between the reference and antibody channel using the cumulative measurement from multiple repeated images.

A strength of the method is that it can be used for particles with a non-spherical topology and uneven staining. It is evident that the signal precision is improved as the distortion of an ovoid bacterium is accounted for through axial normalization in polar coordinates (*Figure 1c*).

**Figure 2.** Validation of site localization method through simulated images with various cell morphologies. (**a**) Examples of simulated images. The different shapes that the method was tested on are shown. From each of the 12 morphologies, 100 time series with 10 frames each were generated. Each frame consists of a set number of photons sampled from the shape's spatial distribution. Before creating a time series, a random translation and rotation were applied to the distribution. The chosen shapes emulate various eccentricities and degrees of surface protein patchiness found, for example, in bacteria as well as the surface irregularity of larger cells. The 1 µm scale bar assumes a pixel length of 20.5 nm corresponding to that of the structured illumination microscopy (SIM) images, with this pixel length the simulated distance between the reference channel and the target channel is 41 nm. (**b**) Site localization results expressed as deviation from simulated distances. The deviations of the measured values from the simulated distances are shown in a violin plot grouped by shape. The mean distance deviations are in order from left to right [mean ± SD, median (IQR)]: 0.2 ± 0.5 nm, 0.1 (0.7) nm; 0.2 (1.0) nm; 0.3 ± 0.6 nm, 0.3 (0.8) nm; 1.1 ± 0.5 nm, 1.0 (0.7) nm; 1.0 ± 0.8 nm, 1.0 (1.1) nm; 1.2 ± 0.8 nm, 1.3 (1.1) nm; 3.0 ± 0.7 nm, 3.1 (0.9) nm; 5.1 ± 1.1 nm, 5.2 (1.3) nm; 2.9 ± 1.0 nm, 2.8 (1.2) nm; 1.0 ± 0.6 nm, 1.0 (0.7) nm; 1.2 ± 0.6 nm, 1.2 (0.8) nm; 1.5 ± 0.8 nm, 1.6 (1.1) nm. (**c**) Bootstrap of measured mean distances as a function of the number of time series used. The plots show an estimate of the 95% confidence interval of the mean of a number of measurements computed by percentile bootstrap with 50,000 resamples. As the number of measurements used in the calculation of the mean increases, the confidence interval narrows.

The online version of this article includes the following figure supplement(s) for figure 2:

**Figure supplement 1.** Site localization results on simulated images expressed as deviation from simulated distance for different types of labelling and signal.

## Site localization measurements on simulated images with various cell morphologies

To assess the performance of the site localization method, we generated simulated fluorescence images with variable surface staining and morphology. The images consisted of two channels created by sampling a fixed number of 'photons' from a spatial distribution. In the reference channel, the distribution was designed to emulate the spatial distribution of photons in a microscope for a cellular surface. In the target channel, a distribution based on the set of points in the reference channel at a chosen perpendicular distance was used. The perpendicular distance between the reference channel distribution and the target channel distribution was chosen to be 2 px corresponding to 41 nm using the pixel length of our N-SIM microscopy setup. In order to mimic the experimental process of acquiring repeated frames of each data point in a time series, for each chosen shape 100 time series were created with 10 frames each. The results, shown in *Figure 2*, indicate that a high accuracy can be achieved for various cell surface patterns. However, for certain shapes, the method yields a slightly larger distance than the true value (*Figure 2b*). The reported accuracy for site localization measurements on the simulations in relation to number of cells is assessed and presented in *Figure 2c*. To explore sources of uncertainty in site localization measurements, additional images were simulated to represent different types of labelling (*Figure 2—figure supplement 1*). As expected, the uncertainty is larger with secondary antibody labelling than with direct labelling. The site localization uncertainty also increases when the SNR is decreased. These results demonstrate the robustness of the site localization method for varying surface expression and morphology.

## Validation of site localization method by measurement of DNA probes on cellular surfaces

In order to test the method in vitro, we performed site localization measurements on surface-tethered dsDNA probes of known lengths. Human erythrocytes were stained with membrane dye CellBrite Fix 488 and Alexa Fluor 647-conjugated dsDNA of varying length (*Figure 3b*). Images were acquired using an N-SIM microscope. To begin with, swelled spherical erythrocytes were analysed with the site localization method as well as with an existing method for determining molecular heights on spherical particles (*Figure 3—figure supplement 1*). The results show good agreement. Site localization was thereafter performed on non-spherical erythrocytes and show results equivalent to that of spherical cells (*Figure 3c*). We further tested the method's capabilities by measuring the height of dsDNA with varying lengths on the surface of non-spherical red blood cells (*Figure 3e*). The measured heights are approximately half of the full length of the DNA strands. The measurements agree well with the predicted worm-like chain (WLC) model height for DNA strands, with a persistence length of 50 nm (*Baumann et al., 1997*), tethered to a surface and able to freely rotate. Altogether, we show that our method can accurately estimate molecular heights on spherical and non-spherical cellular surfaces.

## Site localization measurement of binding sites on bacterial surface protein

Site localization measurements that were performed for ligands on bacterial M protein agree well with previously reported data. The antibody Fc binding site on M protein is located at the S region (*Akesson et al., 1994*), and a non-specific monoclonal IgG antibody (Xolair) is used for site localization of this binding. Additionally, a measurement is carried out for an M protein-specific mAb, Ab49 (*Bahnan et al., 2021*), with an epitope located in the B3-S region, that is, slightly further along the IgGFc binding region. For determining binding to the far end of M protein, we used fibrinogen that has two binding sites at the B1 and B2 region (*Hauri et al., 2019*). We thus expect the binding sites to be arranged in accordance with the schematic shown in *Figure 4—figure supplement 1c*. *S. pyogenes* strain SF370 was stained with Alexa Fluor 488-conjugated WGA and coated with Alexa Fluor 647-conjugated Xolair, Ab49, or fibrinogen (*Figure 4a*). Images of single bacteria were acquired using an N-SIM microscope. The median resolved distance is [median (IQR)] 5.6 (29.7) nm for the Xolair Fc binding site, 6.2 (29.8) nm for Ab49 Fab binding site, and 18.0 (49.3) nm for the two fibrinogen binding sites. The reported accuracy for site localization measurements on *S. pyogenes* in relation to number of biological replicates is assessed and presented in *Figure 4—figure supplement 1*. To further explore sources of uncertainty in site localization measurements, the antibody binding sites were measured with secondary antibody labelling and compared to direct labelling (*Figure 4—figure*

**Figure 3.** Site localization measurements of DNA probes on human red blood cells. (**a**) A schematic illustrating the binding of DNA probe to red blood cells. Human red blood cells were coated with Alexa Fluor 647-conjugated dsDNA of varying lengths and a cholesterol anchor. (**b**) Representative structured illumination microscopy (SIM) images of spherical and non-spherical erythrocytes. Human erythrocytes were stained with membrane dye CellBrite Fix 488 and Alexa Fluor 647-conjugated dsDNA of length 48 bp. Scale bar is 2 μm. (**c**) Site localization method performs well for both spherical and non-spherical particles. The measured distances for 48 bp dsDNA probes on erythrocytes are shown in a violin plot. The resolved distance is [median (IQR), mean ± SD] 11.5 (11.6) nm, 11.4 ± 9.3 nm (n = 31) for DNA on the spherical cells and 9.2 (10.7) nm, 9.8 ± 8.1 nm (n = 33) for the non-spherical erythrocytes. Error bars indicate interquartile range. Additionally, the result for spherical cells was confirmed using an existing method for site localization on spherical objects (*Figure 3—figure supplement 1*). (**d**) Representative SIM images of DNA-coated erythrocytes. Human erythrocytes were stained with membrane dye CellBrite Fix 488 and Alexa Fluor 647-conjugated dsDNA of length 18 bp, 48 bp, and 63 bp. Scale bar is 2 μm. (**e**) Site localization measurements of DNA probes show good agreement with predicted worm-like chain (WLC) heights The measured heights of the DNA probes are plotted as a function of the fully extended length of the DNA strands. The measured height is [median (IQR), mean ± SD] 3.7 (8.8) nm, 5.1 ± 8.3 nm (n = 25) for 18 bp DNA, 9.2 (10.7) nm, 9.8 ± 8.1 nm (n = 33) for 48 bp DNA, and 12.5 (8.2)nm, 10.3 ± 10.5 nm (n = 30) for 63 bp DNA. Error bars indicate interquartile range. The dashed line represents predicted height based on modelling the DNA strand as a WLC with a persistence length of 50 nm free to move in a hemisphere above the surface.

The online version of this article includes the following figure supplement(s) for figure 3:

**Figure supplement 1.** Distance measurements with site localization and cell surface optical profilometry (CSOP) for DNA probes on human red blood cells The measured heights of 48 bp dsDNA probes on erythrocytes are shown in a violin plot.

**Figure supplement 2.** Control of site localization method using double-stain wall control.

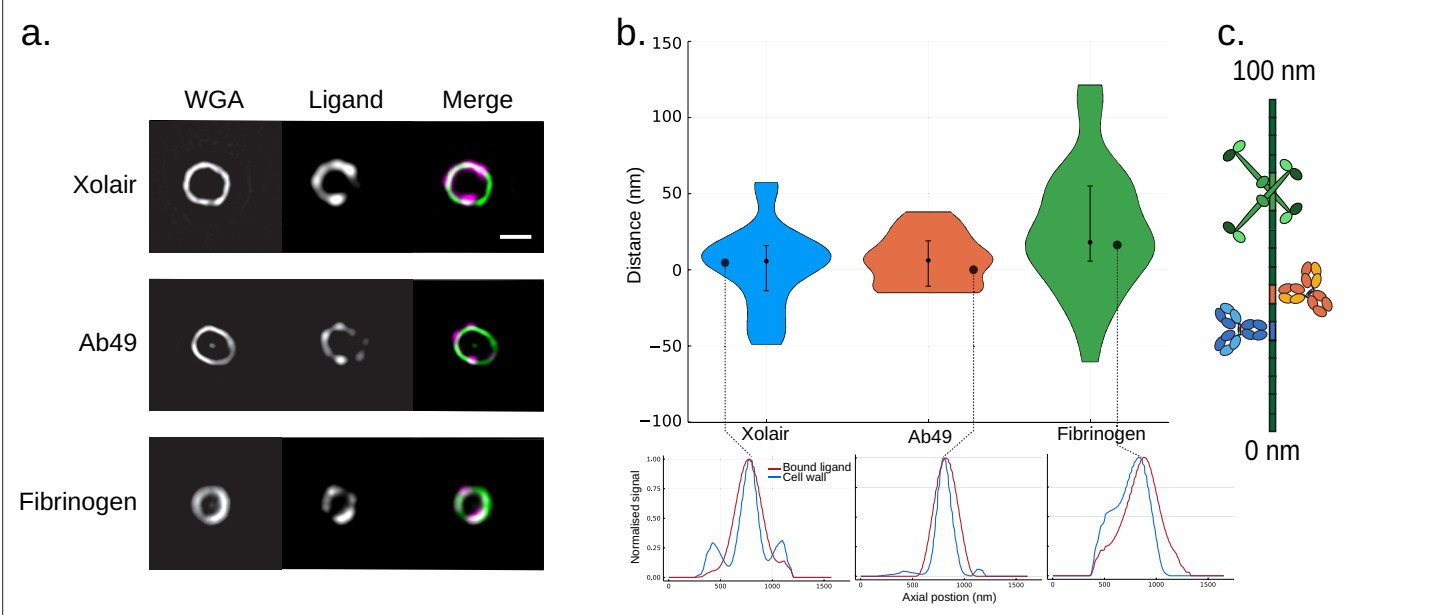

**Figure 4.** Site localization measurements of bound ligands on bacterial surface protein. (**a**) Representative structured illumination microscopy (SIM) images of ligand coated bacteria. *S. pyogenes* strain SF370 has been fixed, stained with Alexa Fluor 488-conjugated wheat germ agglutinin (WGA), and coated with Alexa Fluor 647-conjugated Xolair (top), Ab49 (middle), or fibrinogen. Scale bar is 500 nm. (**b**) Binding site measurements of antibodies and fibrinogen on bacterial M protein The measured distances are shown in a violin plot. The resolved distance between the two channels is [median (IQR), mean ± SD] 5.6 (29.7) nm, 2.5 ± 26.7 nm (n = 41) for the Xolair Fc binding site, 6.2 (29.8) nm, 6.5 ± 16.8 nm (n = 28) for Ab49 Fab binding site, and 18.0 (49.3) nm, 26.2 ± 41.6 nm (n = 33) for the two fibrinogen binding sites. Error bars indicate the interquartile range. An intensity profile from a single bacterium is shown for each of the ligands below the violin plot. (**c**) A schematic illustrating the determined binding sites on M protein.

The online version of this article includes the following figure supplement(s) for figure 4:

**Figure supplement 1.** Percentile bootstrap on site localization results for double wall stain control.

**Figure supplement 2.** A comparison of site localization measurements of antibody binding sites using secondary antibody fragments and direct conjugation.

*supplement 2*). The presented results are consistent with the arrangement of the binding sites as reported in the literature (*Akesson et al., 1994*; *Bahnan et al., 2021*; *Hauri et al., 2019*).

## Site localization measurements using widefield and deconvolved images

By performing site localization measurements on widefield images, we show that our method can yield viable results even with conventional microscopy data. Moreover, deconvolution of the widefield images may yield an increase in precision. Widefield images were acquired with the same optical system (see SIM in previous section), with the exception of the light source used being LED-based instead of laser-based. Comparison of site localization measurements using SIM and widefield was performed on two sets of samples; fibrinogen- and Xolair-coated *S. pyogenes*. Widefield and SIM images were acquired on separate datasets. For the fibrinogen samples (*Figure 5b*), the median resolved distance between the two channels is [median (IQR)] 18.0 (49.3) nm with SIM data, 38.5 (71.0) nm with widefield data, and 40.1 (42.0) nm with deconvolved widefield data. For the Xolair samples (*Figure 5d*), the median resolved distance for the Xolair Fc binding site is 5.6 (29.7) nm with SIM data, 12.3 (76.5) nm with widefield data, and 14.2 (34.5) nm with deconvolved widefield data. This indicates that the site localization method performs well with widefield images and that the precision can be increased through deconvolution.

## Discussion

The locations of protein binding sites on cellular surfaces are important for a wide range of cellular processes. In particular, how and where an antibody binds to pathogen surface structures is critical

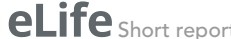

**Figure 5.** A comparison of site localization measurements using structured illumination microscopy (SIM), widefield, and deconvolved widefield images. (**a**) Representative images of Xolair-coated bacteriaare shown at the top. Scale bar is 500 nm. *S. pyogenes* strain SF370 has been fixed, stained with Alexa Fluor 488-conjugated wheat germ agglutinin (WGA), and coated with Alexa Fluor 647-conjugated Xolair antibodies. (**b**) Site localization measurement of Fc binding site yields similar results with the widefield dataset. The resolved distance between the two channels is [median (IQR), mean ± SD] 5.6 (29.7) nm, 2.5 ± 26.7 nm (n = 41) with SIM data, 12.3 (76.5) nm, 8.7 ± 36.6 nm (n = 32) with widefield data, and 14.2 (34.5) nm, 17.0 ± 25.4 nm (n = 31) with deconvolved widefield data. Error bars indicate the interquartile range. (**c**) Representative images of fibrinogen-coated bacteria are shown at the top. *S. pyogenes* strain SF370 has been fixed, stained with Alexa Fluor 488-conjugated WGA, and coated with antibody Alexa Fluor 647-conjugated fibrinogen. Scale bar is 500 nm. (**d**) Site localization measurement of fibrinogen binding sites yields similar results with the widefield dataset. The resolved distance between the two channels is [median (IQR), mean ± SD] 18.0 (49.3) nm, 26.2 ± 41.6 nm (n = 33) with SIM data, 38.5 (71.0) nm, 35.9 ± 53.3 nm (n = 37) with widefield data, and 40.1 (42.0) nm, 41.1 ± 40.1 nm (n = 29) with deconvolved widefield data. Error bars indicate the interquartile range.

The online version of this article includes the following figure supplement(s) for figure 5:

**Figure supplement 1.** Control of site localization method using double-stain wall control.

for the outcome of host-pathogen interactions. This is especially relevant for bacteria as they are known to target antibodies in many different ways (*Nordenfelt et al., 2012*; *Nordenfelt and Björck, 2013*). Knowing the location of the epitope greatly aids both mechanistic understanding and facilitates potential therapeutic development.

We have developed a method for imaging-based localization of binding sites on cellular surface proteins with nanometer-scale precision. Super-resolution techniques such as iPALM, STORM, and STED can in practice achieve an ~10 nm resolution but are costly imaging systems that typically require complex sample preparation. Here, an ~10 nm (~12 nm IQR, ~9 nm SD) precision is achieved with diffraction-limited microscopy images by dimension reduction of surface data within an image plane. Existing imaging-based averaging methods are designed for identical or spherical particles and therefore not suited for site localization on native cell surface morphology. This is made possible with our method through normalization of the axial positions of a reference surface.

To assess the performance of the site localization method, we analysed simulated images with variable staining and morphology. These results indicate that a 1 nm precision could be achievable under ideal conditions and show that the site localization method is highly accurate for varying topologies and inhomogeneous surface expression. To test our method in vitro, we measured the height of DNA probes on erythrocyte surfaces. These results validate that our method can accurately estimate molecular heights on spherical and non-spherical cells. Additionally, the results for spherical cells were confirmed using an existing method for site localization on spherical objects, showing good agreement. We have applied our site localization method to determine known binding site locales on M protein that is unevenly expressed on the bacterial surface of *S. pyogenes*. Our presented results show that this method is suited even for cells with non-spherical topology. However, it has proven difficult to perform site localization measurements on bacteria. This may be due to the small size of bacteria. We show that our method yields viable results with conventional microscopy and demonstrate that the precision may be increased by deconvolution of widefield images. Furthermore, our in silico experiments, as well as our in vitro experiments, indicate that the variation in the measurements can be minimized by increasing SNR and minimizing the biological variability, for example, by using directly conjugated antibodies instead of secondary labelling. We believe this method may be useful for rapid screening of epitope locales, and the implementation, written in the open-source language Julia, is provided on GitHub.

## Materials and methods

**Key resources table**

| Reagent type (species) or resource | Designation | Source or reference | Identifiers | Additional information |
|---|---|---|---|---|
| Strain, strain background (*Streptococcus pyogenes*) | SF370 | ATCC | Cat# 700294 | |
| Biological sample (*Homo sapiens*) | Venous blood (normal, adult) | This paper | | Freshly taken from *Homo sapiens* |
| Antibody | Xolair (humanized from mouse monoclonal) | Novartis | Cat# 028268; RRID#:AB_2459636 | (500 µg/ml) |
| Antibody | Anti-streptococcal M protein; Ab49 (human monoclonal) | *Bahnan et al., 2021* | | (10–µg/ml) |
| Antibody | Alexa Fluor 647 AffiniPure F(ab')₂ fragment goat anti-human IgG (Fab specific) (goat polyclonal) | Jackson ImmunoResearch Laboratories | Cat# 109-606-097; RRID:AB_2337898 | (3 µg/ml) |
| Antibody | Alexa Fluor 647 AffiniPure F(ab')₂ fragment goat anti-human IgG (Fc specific) (goat polyclonal) | Jackson ImmunoResearch Laboratories | Cat# 109-606-170; RRID:AB_2337902 | (3 µg/ml) |
| Other | Alexa Fluor 647 (Invitrogen) | Thermo Fisher | Cat# 820006 | |
| Other | CellBrite Fix Membrane stain 488 | | Cat# 30090 | |
| Sequence-based reagent | Conjugated DNA_F 18 base pair | This paper, *Son et al., 2020* | Oligonucleotide | 5'-Alexa647-N-AGCTGCGGTCAGATC-3' |

*Continued on next page*

*Continued*

| Reagent type (species) or resource | Designation | Source or reference | Identifiers | Additional information |
|---|---|---|---|---|
| Sequence-based reagent | Conjugated DNA_R 18 base pair | This paper, *Son et al., 2020* | Oligonucleotide | 5'-Cholesterol-TEG-TCGACGCCAGTCTAG-3' |
| Sequence-based reagent | Conjugated DNA_F 48 base pair | This paper, *Son et al., 2020* | Oligonucleotide | 5'-Alexa647-N-GATCTGACCGCAGCTATCT |
| Sequence-based reagent | Conjugated DNA_R 48 base pair | This paper, *Son et al., 2020* | Oligonucleotide | 5'-Cholesterol-TEG-CTAGACTGGCGTCGATA |
| Sequence-based reagent | Conjugated DNA_F 63 base pair | This paper, *Son et al., 2020* | Oligonucleotide | GACGTACTGCTATTGCTA GCGATTCCAT- |
| Sequence-based reagent | Conjugated DNA_R 63 base pair | This paper, *Son et al., 2020* | Oligonucleotide | 5'-Cholesterol-TEG-ATGGAATCGCTAGC AATAGCAGTACGTCTACA TGAAGTCGTTGATTCGAC GCCAGTCTAGTAG-3' |
| Peptide, recombinant protein | Fibrinogen from human plasma, Alexa Fluor 647 | Thermo Fisher | Cat# F35200 | (20 μg/ml) |
| Peptide, recombinant protein | Wheat germ agglutinin, Alexa Fluor 488 | Thermo Fisher | Cat# W11261 | (1:250) |
| Peptide, recombinant protein | Wheat germ agglutinin, Alexa Fluor 647 | Thermo Fisher | Cat# W32466 | (1:250) |
| Peptide, recombinant protein | IdeS | Hansa Biopharma | | |
| Software, algorithm | FluoroDist | This paper, GitHub | https://github.com/nordenfeltLab/FluoroDist.jl (*Kumra Ahnlide, 2022*; copy archived at swh:1:rev:053e516413277cf488ef9db1a24ce576e24d18f9) | Simulated fluorescence images with accurate surface distances |
| Software, algorithm | RegisterQD | GitHub, *Greer, 2019* | https://github.com/Holylab/RegisterQD.jl | Image registration with the QuadDIRECT optimization algorithm |
| Software, algorithm | Site Localization | This paper, GitHub | https://github.com/nordenfeltLab/SiteLocalization | Binding site localization on non-homogeneous cell surfaces using topological image averaging |

## Bacterial culturing conditions

*S. pyogenes* strain SF370 wildtype was cultured overnight in THY medium (Todd Hewitt Broth; Bacto; BD, complemented with 0.2% [w/v] yeast) at 37°C in an atmosphere supplemented with 5% $CO_2$. Strain SF370 expresses M1 protein on its surface and is available through the American Type Culture Collection (ATCC 700294) (*Ferretti et al., 2001*). The bacteria were harvested at early log phase and washed twice with PBS.

## Opsonization of bacteria

Xolair (omalizumab, Novartis) is a humanized monoclonal IgG that is IgE-specific, and thus only binds to M protein via Fc. Ab49 is an M protein-specific antibody (*Bahnan et al., 2021*). For secondary antibody labelling, both antibodies were treated with IdeS (Hansa Biopharma) (*von Pawel-Rammingen et al., 2002*), an enzyme that cleaves IgG at the hinge region, separating the F(ab')$_2$ from the Fc. The fibrinogen used here was isolated from human plasma and conjugated with Alexa Fluor 647 (Invitrogen).

## Antibody conjugation

Direct conjugation of antibodies was done using the fluorescent dye Alexa Fluor 647 (Invitrogen). The desiccated dye was dissolved in DMSO at a concentration of 10 mg/ml. The antibodies were

concentrated to a concentration of 2 mg/ml, and 1/10th of the final volume of sodium bicarbonate (1 M, pH 8.3) was added. The dye was added at a final concentration of 75 μg/mg of antibody. The antibodies were then incubated with the dye at room temperature (RT) for an hour. To remove any unbound dye, the antibodies were transferred to 50 kDa Amicon ultra filter columns (Merck Millipore). The antibodies were washed by adding PBS and centrifuging (12,000 × $g$). This was done three times to ensure all unbound dye had been removed. Conjugated antibody concentration and degree of labelling were assessed by using a DeNovix DS-11 FX spectrometer.

### Fixation and staining of bacteria
Bacteria were sonicated (VialTweeter; Hielscher) for 0.5 min to separate any aggregates and incubated fixed in 4% paraformaldehyde for 5 min on ice. The bacteria were thereafter washed with PBS twice (10,000 × $g$, 2 min). SF370 wildtype was stained with Alexa Fluor 488-conjugated WGA. Bacteria were incubated with IdeS-cleaved Xolair, Ab49, or Alexa Fluor 647-conjugated Fibrinogen (Invitrogen). The antibody samples were stained with fluorescently labelled IgGFab- or IgGFc-specific F(ab')$_2$ fragments (Alexa Fluor 647-conjugated anti-human IgGFc or IgGFab; Jackson ImmunoResearch Laboratories). Samples were set on glass slides using ProLong Gold Antifade Mountant with No. 1.5 coverslips.

### Red blood cell measurements
DNA height probes were synthesized by Integrated DNA Technologies and purified with HPLC and resuspended in Tris buffer with 1 mM EDTA. The two strands are composed of a forward oligonucleotide and a reverse oligonucleotide. The sequences can be found in Key resources table. Venous blood was taken from a human donor and washed with PBS three times (750 × $g$, 5 min). The red blood cell pellet was treated with TrypLE at 37°C for 15 min and thereafter washed once (750 × $g$, 5 min). The red blood cells were then resuspended in CellBrite Fix 488 membrane staining solution (1:200 of stock solution 1000X) and incubated for 15 min at RT. For swelling of cells, the staining was in a 70% isotonic solution. The cells were fixed in a solution of 4% paraformaldehyde and 0.2% glutaraldehyde at RT for 20 min. The cells were then washed twice (500 × $g$, 5 min) and resuspended in 500 μl PBS. 1 μl of DNA height probe was added to 3 μl of fixed red blood cells in 50 μl PBS. Samples were set on glass slides using ProLong Gold Antifade Mountant with No. 1.5 coverslips.

### DNA height prediction
The DNA height prediction was performed by using a previously derived analytical expression for the probability distribution ($p(r,t)$) of the end-to-end distance in the WLC model (***Murphy et al., 2004***):

$$p(r,t) = \frac{4\pi A(t) r^2}{(1-r^2)^{9/2}} \exp\left(\frac{-3t}{4(1-r^2)}\right)$$

where A is a normalization constant defined as

$$A(t) = \frac{4\left(\frac{3t}{4}\right)^{3/2} \exp\left(\frac{3t}{4}\right)}{\pi^{3/2}\left(4 + \frac{12}{3t/4} + \frac{15}{(3t/4)^2}\right)}$$

and $r$ and $t$ are defined as

$$r = \frac{L}{L_p}, t = \frac{R}{L}$$

where $L$ is the contour length, $L_p$ is the persistence length, and $R$ is the end-to-end distance. The expected end-to-end length given a persistence length of 50 nm was calculated for each DNA strand length. The height was then obtained by finding the centre of mass of a rod with the calculated length tethered to a surface with one end allowed to rotate freely in a hemisphere (***Son et al., 2020***).

### Simulation of fluorescence images
To generate the fluorescence images, plane curves with a parameterization were chosen. The curves that were not ellipses were chosen to be quadratic B-splines since this yields a continuous derivative and makes the necessary computations simple. To get closer to an arc-length parameterization, the curve was discretized by walking around the curve and at each point taking steps inversely proportional to the analytically determined magnitude of the gradient at that point. The photons for the

reference channel were generated by sampling values from this discrete parameterization. To simulate the membrane width, an offset perpendicular to the gradient was sampled from a uniform distribution of the desired width. Another random offset was sampled from a two-dimensional normal distribution to simulate the effect of the point spread function. The target channel was simulated in the same way but with an additional offset in the direction normal to the gradient at the sampled point of the curve. When antibody labelling was simulated, a 15 nm offset was added in a random three-dimensional direction for each antibody. For all included simulations, an offset of 2 px was chosen, corresponding to a distance of 41 nm for a pixel length of 20.5 nm. The code was implemented in Julia and is available on GitHub (nordenfeltLab/FluoroDist.jl).

### SIM image acquisition

Images of single bacteria were acquired using a Nikon N-SIM microscope with LU-NV laser unit, CFI SR HP Apochromat TIRF ×100 Oil objective (N.A. 1.49) and an additional ×1.5 magnification. The camera used was ORCA-Flash 4.0 sCMOS camera (Hamamatsu Photonics K.K.), and the images were reconstructed with Nikon's SIM software on NIS-Elements Ar (NIS-A 6D and N-SIM Analysis). Fluorescent beads (100 nm) were imaged to measure and correct for chromatic aberration, as well as for the N-SIM grating alignment. Single cells were manually identified and imaged with 488 and 640 nm lasers in time series with 15–20 frames, depending on initial sample signal. For CSOP (*Son et al., 2020*) analysis, images of singles cells were acquired in 15 slices with step sizes of 50 nm.

### Microscope calibrations

TetraSpeck 0.1 µm fluorescent microspheres are mounted on No. 1.5 coverslips using ProLong Gold Antifade Mountant in the same manner as the bacterial samples. These beads are used for the objective collar correction, SIM grating alignment, and measurement of SIM and widefield PSF. Images of the beads were acquired and chromatic aberration was corrected for by performing image registration (Holylab/RegisterQD.jl) and applying the found transform to all images.

### Site localization analysis

A Circle Hough Transform (*Illingworth and Kittler, 1987*) and Canny Edge Detection (*Canny, 1986*) were used for circle fitting and edge detection, respectively. Images were filtered by SNR (approximately > 3) as calculated using

$$10 \log_{10} \left( \frac{\max_{\text{image}}}{\text{median}_{\text{image}}} \right)$$

as an SNR estimate. The time series were cut as the SNR relative to the first time frame fell below a given percentage (typically 30%), which is set as an input to the code. A polar transformation of the found circle was performed on a bicubic interpolation of the image. The alignment of the peak intensity was performed by identifying a peak maximum using a sliding average. Labelling of the cell wall with two different fluorescent dyes should give a distance estimation of zero. The measured offset, shown in *Figure 3—figure supplement 2* and *Figure 5—figure supplement 1*, is likely due to chromatic aberration at the imaging plane. This offset is used to correct for chromatic aberration in subsequent site localization measurements by correcting the position of the reference region prior to calculating distance to the ligand signal. To avoid attaining out-of-focus peaks in the ligand channel, peak identification was performed beyond the reference region. The number of bacteria, together with median (IQR) and mean ± SD, is given in the figure captions. The widefield images were deconvolved using the Richardson–Lucy algorithm (*Richardson, 1972*; *Lucy, 1974*) in 10 iterations. The analysis pipeline, written in Julia, is provided on GitHub (nordenfeltLab/SiteLocalization).

## Acknowledgements

We thank Jonas Tegenfeldt and Elke Hebisch for support during the early phase of the method development. We also thank Oscar André for help with the blood assay. *Figure 3a* was created using Biorender.com. We thank Lund University Bioimaging Center (LBIC) for use of microscopes.

## Additional information

### Funding

| Funder | Grant reference number | Author |
|---|---|---|
| Vetenskapsrådet | 2016-01071 | Pontus Nordenfelt |
| Knut och Alice Wallenbergs Stiftelse | 2016.0023 | Pontus Nordenfelt |
| Cancerfonden | 19 0444 Pj | Pontus Nordenfelt |
| Royal Physiographic Society in Lund | 2018 | Vibha Kumra Ahnlide |
| Vetenskapsrådet | 2018-05795 | Pontus Nordenfelt |
| Vetenskapsrådet | 2020-01511 | Pontus Nordenfelt |

The funders had no role in study design, data collection and interpretation, or the decision to submit the work for publication.

### Author contributions

Vibha Kumra Ahnlide, Conceptualization, Data curation, Formal analysis, Investigation, Methodology, Software, Validation, Visualization, Writing - original draft, Writing – review and editing; Johannes Kumra Ahnlide, Formal analysis, Methodology, Software, Validation, Visualization, Writing – review and editing; Sebastian Wrighton, Investigation, Writing – review and editing; Jason P Beech, Investigation, Supervision, Writing – review and editing; Pontus Nordenfelt, Conceptualization, Funding acquisition, Project administration, Resources, Supervision, Writing – review and editing

### Author ORCIDs

Sebastian Wrighton http://orcid.org/0000-0002-3378-7925
Pontus Nordenfelt http://orcid.org/0000-0002-9481-9951

### Ethics

Human subjects: Informed consent was retrieved for the collection of capillary blood from finger pricking.

### Decision letter and Author response

Decision letter https://doi.org/10.7554/eLife.64709.sa1
Author response https://doi.org/10.7554/eLife.64709.sa2

## Additional files

### Supplementary files

• Transparent reporting form

### Data availability

Data has been made available on Dryad. The source code of the software developed in this study is available through Github: https://github.com/nordenfeltLab/SiteLocalization (copy archived at swh:1:rev:de1f827dbf03cd0a77ee5c03a7fd8c747541f7fb); https://github.com/nordenfeltLab/Fluoro-Dist.jl (copy archived at swh:1:rev:053e516413277cf488ef9db1a24ce576e24d18f9).

The following dataset was generated:

| Author(s) | Year | Dataset title | Dataset URL | Database and Identifier |
|---|---|---|---|---|
| Nordenfelt P, Ahnlide KumraV, Kumra Ahnlide J, Wrighton S, Beech J | 2022 | Nanoscale binding site localization by molecular distance estimation on native cell surfaces using topological image aBinding Site Localization by Molecular Distance Estimation on Native Cell Surfaces Using Topological Image Averaging | https://doi.org/10.5061/dryad.h18931znn | Dryad Digital Repository, 10.5061/dryad.h18931znn |

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
