## [Editor Report]

You have developed and validated a new method for measurement of nanoscale height of macromolecules that can be non-uniformly distributed on irregular surfaces. Such samples are common in biology, which will make this a valuable approach to achieve super-resolution results for samples where this would have not been possible previously.

---

## [Decision Letter]

**Decision letter after peer review:**

Thank you for submitting your article "Binding site localization on non-homogeneous cell surfaces using topological image averaging" for consideration by *eLife*. Your article has been reviewed by 3 peer reviewers, including Michael L Dustin as the Reviewing Editor and Reviewer #1 and Erdinc Sezgin as Reviewer #2, and the evaluation has been overseen by Olga Boudker as the Senior Editor.

The reviewers have discussed the reviews with one another and the Reviewing Editor has drafted this decision to help you prepare a revised submission.

Summary:

You describe a method to measure nanoscale height of fluorophors above a surface that can be rendered with a distinct fluorophore, but doesn't need to be radially symmetric, as for recently published CSOP method. This was applied to measurement of *S. pyogenes* M protein.

There were a number of technical concerns about the calibration and validation of the method and the comparison to CSOP.

Essential revisions:

1. Since the major focus of this work is applying the method of spatially averaging of two fluorescent signals to non-spherical surfaces, it is critical that the image analysis method (using Hough transforms, edge detection, filtering, etc) be validated with known non-spherical objects to confirm that it is working as expected. Currently, it is unclear for what range of sizes and shapes the approach is accurate and what potential sources of error might be. Both experiments and simulated data could be used to evaluate performance of the analysis algorithm.

2. The authors indicate they use 100nm beads to correct for chromatic aberration, but chromatic aberrations can increase as the imaging plane moves away from the glass surface. Do the dual-labeling experiments shown in Figure 3 – Supplementary Figure 1 include correction of chromatic aberration based on the 100nm beads? Is so, then it appears that chromatic aberration needs to be corrected at the image plane rather than the glass surface. How is this difference observed in the dual labeling experiment accounted for in the image analysis procedure?

3. How does the reported accuracy depend on the number of cells averaged and the amount of fluorescent antibody on each cell? It would be interesting to see a plot of the height uncertainty per cell as a function of the antibody fluorescence intensity (as a proxy for amount of antibody). Since the reported ~5nm resolution is for averages across 37-55 independent cells, would averaging more cells give higher precision (and does averaging fewer cells give lower precision)?

4. How does fixation affect the measurements? Can the method be used on unfixed cells, and what is the localization uncertainty for unfixed samples? Also, is the uncertainty in localization of the ligand signal due to antibody fluctuations and labeling location included in the analysis?

5. In the comparison with CSOP (Son, PNAS, 2020), it would be helpful to first demonstrate both techniques on spherical objects, in order to confirm that the use of CSOP is consistent with published results and to evaluate how the new method compares for spherical objects (presumably it should be equivalent). There is also not a calibration test in the study. There should be a very well controlled calibration samples to evaluate the accuracy as well as dynamics range. The authors could use swelled spherical RBC with the same DNA based height standards examined in Son et al. in GUV. CSOP and the new method could then be fairly compared, calibrated and then applied to the same height standards on a biconcave RBC, which would test the new method on a non-spherical object.

[Editors' note: further revisions were suggested prior to acceptance, as described below.]

Thank you for submitting your revised article "Nanoscale binding site localization by molecular distance estimation on native cell surfaces using topological image averaging" for consideration by *eLife*. Your article has been re-reviewed by the same 3 peer reviewers as in the initial review, including Michael L Dustin as the Reviewing Editor and Reviewer #1, and the evaluation has been overseen by Olga Boudker as the Senior Editor. The following individual involved in review of your submission has agreed to reveal their identity: Erdinc Sezgin (Reviewer #2).

The reviewers all felt that the efforts you have made are helpful, but still were not clear on the height accuracy of the experimental data needed to be useful for relevant measurements on biological molecules at surfaces,. These revisions may not require new experiments, but will require clarification of the data sets, some revisions to the simulated data shown, and additional analysis of the signal requirements and sources of error. The Reviewing Editor has drafted this to help you prepare a second revised submission.

Essential revisions:

You have provided additional simulations and experimental data in the revised manuscript, but you have not fully addressed the essential revisions as the most significant issue for any new method, the experimental validation, is not complete. Showing that an algorithm can obtain height measurements on simulated data demonstrates the performance of the algorithm but is not true experimental validation of the technique. Specifically, the authors need to confirm that molecules of known heights can be correctly measured with the claimed precision. Furthermore, the reported experimental height uncertainties are significantly different from the simulated uncertainties for variable shapes, which raises fundamental questions about other sources of error.

1. You have not shown that experimental measurements and the analysis algorithm can correctly capture the height of a known molecule (e.g. DNA) on a known surface (e.g. a sphere). The measurement of DNA height on the non-spherical surface of an RBC is not meaningful until you can show that you can accurately measure the height of that DNA on a known surface geometry. It is the equivalence of those two height measurements – one on a known surface and one on a variable surface – that would give confidence that you have a new methodology for obtaining accurate height measurements on non-spherical and non-uniform surfaces. One suggestion was to use a swelled erythrocyte in a 50% isotonic solution as a spherical object, but a more standard spherical object would be a GUV or uniform glass bead. But you need experimental data on spherical object comparing CSOP and Site Localization.

2. The simulations in Figure 2 are a useful addition to the manuscript and provide a quantitative characterization of the algorithm for idealized images, but the results in Figure 2 show uncertainties much less than that of most experimental measurements in the manuscript. For example, when you say the largest deviation from true distance was 10%, its not obvious what data this claim is based on. The worst case seems to be a 5 nm deviation from true distance, does this mean that the simulated structure was 50 nm in height? Percentage of deviations should be clearly shown and explained. Furthermore, what are the dominant sources of localization uncertainty if the contribution from patchy and non-spherical surfaces is normally much smaller than 5nm? The precision of fluorescence localization methods typically depends on the number of photons collected. Is that true for this method?

3. To report uncertainty in your height measurements, you use standard error of the mean rather than standard deviation, which is the more appropriate measure in our collective view. Since uncertainty in height measurement is based on molecular averages on an individual particle basis (affected by, e.g., number of photons collected for molecules on that particle, how thoroughly chromatic aberration is corrected, etc), measuring more particles does not correct those issues for non-random sources of error (e.g. chromatic aberration). It is also unclear to us how removing negative height values from the population of measurements is justified when calculating means if there is no independent reason to reject those measurements as invalid, wouldn't removing only the negative outliers skew the result and to larger errors in the absolute height? Regardless, were you to use standard deviation, rather than standard error of the mean, the reported uncertainty would be significantly larger, which would necessitate a effort to find sources of error and to reduce these as much as possible to make the method useable.

4. While the motivation to measure heights of molecules on patchy, non-spherical surfaces is a good one it should be made clear that CSOP is only valid for spherical objects and it should not be applied to non-spherical objects. There is no reason to show simulated data that CSOP fails on non-spherical objects as applying CSOP to a non-spherical object, even in a simulation, is not appropriate. The important comparison between CSOP and Site Location is that they should both achieve 1 nm measurement accuracy for height measurements on a spherical object, with Site Localization striving to preserve this accuracy on on non-spherical surfaces.

---

## [Author Response]

Essential revisions:1. Since the major focus of this work is applying the method of spatially averaging of two fluorescent signals to non-spherical surfaces, it is critical that the image analysis method (using Hough transforms, edge detection, filtering, etc) be validated with known non-spherical objects to confirm that it is working as expected. Currently, it is unclear for what range of sizes and shapes the approach is accurate and what potential sources of error might be. Both experiments and simulated data could be used to evaluate performance of the analysis algorithm.

We have performed extensive additional validation of our method. As the reviewers propose, we have used both new experimental data in terms of live red blood cell analysis of different shapes, as well as extensive simulations of different morphology and staining patterns. See new Figure 2. and new Figure 5., as well as Figure 3-supplementary note 1. We also made additional comparisons with the CSOP approach. The CSOP approach introduces large errors even with minor deviations from circular shapes (See new Figure 1—figure supplement 2). It is clear that our site localization approach works well for many different shapes (errors often between 0.2-1.0 nm for microscale objects) and that very eccentric, elongated shapes with patchy staining introduces the most errors in height estimation (about 5 nm for our simulated cells).

2. The authors indicate they use 100nm beads to correct for chromatic aberration, but chromatic aberrations can increase as the imaging plane moves away from the glass surface. Do the dual-labeling experiments shown in Figure 3 – Supplementary Figure 1 include correction of chromatic aberration based on the 100nm beads? Is so, then it appears that chromatic aberration needs to be corrected at the image plane rather than the glass surface. How is this difference observed in the dual labeling experiment accounted for in the image analysis procedure?

We have previously performed 2D chromatic aberration correction by measuring 100 nm beads at the glass surface, which was minimal. However, the reviewers are correct in that we appear to have additional chromatic aberration on the imaging plane, as can be seen in Figure 3—figure supplement 1. This has affected the approximation of absolute position, but not relative. We have decided to correct all data with this measured offset to approximate absolute position. This is done similarly to how Son et al. have corrected for chromatic aberration by measuring dual-stained beads.

3. How does the reported accuracy depend on the number of cells averaged and the amount of fluorescent antibody on each cell? It would be interesting to see a plot of the height uncertainty per cell as a function of the antibody fluorescence intensity (as a proxy for amount of antibody). Since the reported ~5nm resolution is for averages across 37-55 independent cells, would averaging more cells give higher precision (and does averaging fewer cells give lower precision)?

Having enough fluorescent intensity relative to the background is important to be able to get a height estimate. In the algorithm we always calculate a signal to noise ratio for each image. Since we do multiple measurements of the same field of view, we will also bleach the sample and reduce the SNR for each consecutive acquisition. In that sense, the method will always generate fluorescent intensity dependency data for every image analyzed, including where the boundary is for a good height estimate. The absolute boundary is variable, and can be determined by looking at the data, but we only account for measurements that have at least an SNR of 3, as calculated using our SNR estimate included in the provided code, in the first frame. We have added this information to the methods section.

The reported accuracy for site localization measurements in relation to number of cells is assessed and presented both for simulations (Figure 2) and experiments with *S. pyogenes* (Figure 4—figure supplement 1).

We have performed bootstrap analysis of the measurements as a function of replicates. This gives us an estimate of the 95% confidence interval of the mean of a number of measurements computed by percentile bootstrap. The results show at what rate and to what degree the confidence interval narrows as the number of measurements used in the calculation of the mean is increased. In general, 5-10 measurements are typically enough to yield a good height estimate, and more than 20 measurements only gives a small increase in certainty.

4. How does fixation affect the measurements? Can the method be used on unfixed cells, and what is the localization uncertainty for unfixed samples? Also, is the uncertainty in localization of the ligand signal due to antibody fluctuations and labeling location included in the analysis?

This is a good point that we expected to make a difference, but never got around to investigate. Fixation should not alter the surface protein structure, and antibodies typically bind well also to fixed proteins. However, especially the SIM imaging quality is decreased with live cells due to movement as SIM requires 15 images to generate one reconstructed SIM image. We have now tested the performance of the site localization method with live red blood cells. We exploit the fact that we have different shapes in the samples and combine that with the information from the new simulated data in Figure 2. Through this we could infer an expected error due to shape (up to 5 nm), and then we could compare that with the reported error (around 10 nm). It is difficult to assess the exact contribution of all biological and imaging factors, but it appears that live imaging could introduce an additional 5 nm/doubling of error in height estimate. Molecular fluctuations are not considered or accounted for in the analysis.

5. In the comparison with CSOP (Son, PNAS, 2020), it would be helpful to first demonstrate both techniques on spherical objects, in order to confirm that the use of CSOP is consistent with published results and to evaluate how the new method compares for spherical objects (presumably it should be equivalent). There is also not a calibration test in the study. There should be a very well controlled calibration samples to evaluate the accuracy as well as dynamics range. The authors could use swelled spherical RBC with the same DNA based height standards examined in Son et al. in GUV. CSOP and the new method could then be fairly compared, calibrated and then applied to the same height standards on a biconcave RBC, which would test the new method on a non-spherical object.

The reviewers make several good points here, which we have addressed with both experiments and simulations. We have made additional comparisons with the CSOP approach, and this is presented as a supplemental figure (see Figure 1—figure supplement 2). We find that CSOP performs well with spherical objects but not so well with objects with higher eccentricity and patchiness.

We really liked the idea of the red blood cell experiments and tried to do measurements with similar DNA height standards as Son et al. did. We established a method to do measurements directly in blood, and we were able to acquire images of high quality. However, it was apparent that most of the DNA probes interacted with the RBCs in unexpected ways. Some seemed to get stuck in the glycocalyx creating high intensity clusters on the cell surfaces, and others appeared to get internalized. GUVs are likely much more uniform, and have limited other interacting factors, but we wanted to try the method under as difficult conditions as possible. One of the DNA probes seemed to yield consistent behavior, so we acquired data using that to evaluate performance on live samples of different shapes. The DNA binding data showed that live imaging of different shapes introduced a slightly larger error (5-10 nm total error) than simulations (which would constitute perfect conditions). See Figure 5.

We agree that well-controlled calibration samples are important for careful evaluation of the method. For this purpose, we developed an algorithm to simulate images with known characteristics, and then used that to generate differently shaped and stained samples. The site localization algorithm were run on these images and allowed us to benchmark the method carefully, and pinpoint the limitations and strengths of our approach. See Figure 2. See also response to point 1.

References:

Son, S. et al. Molecular height measurement by cell surface optical profilometry (CSOP). Proc Natl Acad Sci USA 117, 14209–14219 (2020).

Raz, A. and Fischetti, V. A. Sortase A localizes to distinct foci on the *Streptococcus pyogenes* membrane. Proc Natl Acad Sci USA 105, 18549–18554 (2008).

Raz, A. et al. *Streptococcus pyogenes* Sortase Mutants Are Highly Susceptible to Killing by Host Factors Due to Aberrant Envelope Physiology. PLoS ONE 10, e0140784 (2015).

[Editors' note: further revisions were suggested prior to acceptance, as described below.]

Essential revisions:You have provided additional simulations and experimental data in the revised manuscript, but you have not fully addressed the essential revisions as the most significant issue for any new method, the experimental validation, is not complete. Showing that an algorithm can obtain height measurements on simulated data demonstrates the performance of the algorithm but is not true experimental validation of the technique. Specifically, the authors need to confirm that molecules of known heights can be correctly measured with the claimed precision. Furthermore, the reported experimental height uncertainties are significantly different from the simulated uncertainties for variable shapes, which raises fundamental questions about other sources of error.1. You have not shown that experimental measurements and the analysis algorithm can correctly capture the height of a known molecule (e.g. DNA) on a known surface (e.g. a sphere). The measurement of DNA height on the non-spherical surface of an RBC is not meaningful until you can show that you can accurately measure the height of that DNA on a known surface geometry. It is the equivalence of those two height measurements – one on a known surface and one on a variable surface – that would give confidence that you have a new methodology for obtaining accurate height measurements on non-spherical and non-uniform surfaces. One suggestion was to use a swelled erythrocyte in a 50% isotonic solution as a spherical object, but a more standard spherical object would be a GUV or uniform glass bead. But you need experimental data on spherical object comparing CSOP and Site Localization.

We understand the reviewers' reasoning in that we need to validate the method experimentally for a known distance measurement. Therefore, we have performed high resolution imaging of DNA standards on fixed and swelled RBC. We have acquired data on spherical cells for comparing CSOP and Site Localization (Figure 3- Supplement 1) showing similar results and precision. Additionally, we show similar results with site localization for spherical and non-spherical RBC (Figure 3), thus validating that our method works for non-spherical cells.

We also previously validated and tested the method for nonuniform surfaces in the simulations (Figure 2), and through experiment with the bacteria samples. All data has been updated with the new considerations regarding uncertainty, and we report median with IQR by default.The bacteria have non-uniform expression of M protein and the relative binding sites are consistent with other reports.

We have had issues using CSOP. We have taken images with the same type of DNA standards in Z-stacks as described in their paper. However, we had issues getting reasonable results due to fitting and segmentation. After familiarising ourselves with all the parts of the CSOP analysis, we realised that we had to manually exclude the majority of our cells since CSOP is highly sensitive to single outliers in a z-stack. This exclusion is based on whether CSOP managed to find the cell in the majority of the stacks, and whether the axial circle found by CSOP fit reasonably to the data. In Author response image 1 we show an example of an included cell as well as an excluded cell for CSOP together with the relevant output from CSOP. In comparison, we did not have to manually curate raw data points for our SiteLoc algorithm. At least in our hands, SiteLoc, is much more robust. We do not include this in the manuscript, only that we achieve similar results for circular objects (Figure 3 – supplement 1).

**Author response image 1. sa2fig1:** Example of data selection for CSOP analysis.

Finally, we further tested the method's capabilities by measuring the height of dsDNA with varying lengths on the surface of fixed red blood cells. These results show a good agreement to the predicted length of DNA strands tethered to a surface and able to move freely (Figure 3e).

2. The simulations in Figure 2 are a useful addition to the manuscript and provide a quantitative characterization of the algorithm for idealized images, but the results in Figure 2 show uncertainties much less than that of most experimental measurements in the manuscript. For example, when you say the largest deviation from true distance was 10%, its not obvious what data this claim is based on. The worst case seems to be a 5 nm deviation from true distance, does this mean that the simulated structure was 50 nm in height? Percentage of deviations should be clearly shown and explained. Furthermore, what are the dominant sources of localization uncertainty if the contribution from patchy and non-spherical surfaces is normally much smaller than 5nm? The precision of fluorescence localization methods typically depends on the number of photons collected. Is that true for this method?

Indeed, the simulated structure was 41 nm in height. However, we have chosen to omit the reference to deviation as a percentage. As the comment notes, keeping the deviation as a ratio would require further exposition and we do not believe that this would benefit the reader since the error isn’t proportional to the height of the structure.

The reviewers are right to note that precision of the method is affected by the number of photons collected. Since we wanted to test specifically that the method is capable of handling other shapes and patchy expression we chose to keep the total number of collected photons constant. In most real cases, patchy expression leads to lower total signal in addition to the difficulty of summing up the signal in a useful way. To better characterize the possible sources of error we added Figure 2—figure supplement 1 where we simulate an experiment using secondary antibody labeling, which would better resemble the experimental conditions originally presented in the manuscript, as well as lower signal (10^4^ ”photons” sampled instead of 10^5^).

3. To report uncertainty in your height measurements, you use standard error of the mean rather than standard deviation, which is the more appropriate measure in our collective view. Since uncertainty in height measurement is based on molecular averages on an individual particle basis (affected by, e.g., number of photons collected for molecules on that particle, how thoroughly chromatic aberration is corrected, etc), measuring more particles does not correct those issues for non-random sources of error (e.g. chromatic aberration). It is also unclear to us how removing negative height values from the population of measurements is justified when calculating means if there is no independent reason to reject those measurements as invalid, wouldn't removing only the negative outliers skew the result and to larger errors in the absolute height? Regardless, were you to use standard deviation, rather than standard error of the mean, the reported uncertainty would be significantly larger, which would necessitate a effort to find sources of error and to reduce these as much as possible to make the method useable.

We understand the reviewer's remark. Since many of the datasets we have acquired have a non-gaussian appearance, we have now decided to use the median of the population as the reported length estimate and thereof the interquartile range as the uncertainty of the measurements. (Figure 2,3,4,5).

Negative outliers were not removed from the datasets. To avoid obtaining out-of-focus peaks in the ligand channel, peak identification was performed beyond the reference region. This is why we did not acquire any negative values.

However, after considering the reviewers comments regarding how this can skew the data distribution, we have now not applied the function of peak identification beyond the reference region.

To identify sources of uncertainty and reduce these, we have performed in silico as well as in vitro experiments. These indicate that the variation in the measurements can be minimized by increasing the SNR and minimizing the biological variability by e.g. using directly conjugated antibodies instead of secondary labelling (Figure 2- supplement 1, Figure 4- supplement 1).

4. While the motivation to measure heights of molecules on patchy, non-spherical surfaces is a good one it should be made clear that CSOP is only valid for spherical objects and it should not be applied to non-spherical objects. There is no reason to show simulated data that CSOP fails on non-spherical objects as applying CSOP to a non-spherical object, even in a simulation, is not appropriate. The important comparison between CSOP and Site Location is that they should both achieve 1 nm measurement accuracy for height measurements on a spherical object, with Site Localization striving to preserve this accuracy on on non-spherical surfaces.

We understand the reviewers comment and have removed the figures showing results with CSOP on non-spherical and non-uniform cells. We now only use CSOP when we compare height measurement on perfectly spherical RBCs.